# Evaluation of the Failure Mechanism in Polyamide Nanofibre Veil Toughened Hybrid Carbon/Glass Fibre Composites

**DOI:** 10.3390/ma15248877

**Published:** 2022-12-12

**Authors:** Ashley Blythe, Bronwyn Fox, Mostafa Nikzad, Boris Eisenbart, Boon Xian Chai, Patrick Blanchard, Jeffrey Dahl

**Affiliations:** 1Department of Mechanical and Product Design Engineering, Swinburne University of Technology, Hawthorn, VIC 3122, Australia; 2Research and Innovation Center, Ford Motor Company, Dearborn, MI 48120, USA

**Keywords:** carbon/glass hybrid composites, stacking sequence, FRP laminates, interlayer hybrid

## Abstract

The interface of hybrid carbon/E-glass fibres composite is interlayered with Xantu.layr^®^ polyamide 6,6 nanofibre veil to localise cracking to promote a gradual failure. The pseudo-ductile response of these novel stacking sequences examined under quasi-static three-point bending show a change to the failure mechanism. The change in failure mechanism due to the interfacial toughening is examined via SEM micrographs. The incorporation of veil toughening led to a change in the dominant failure mechanism, resulting in fibre yielding by localised kinking and reduced instances of buckling failure. In alternated carbon and glass fibre samples with glass fibre undertaking compression, a pseudo-ductile response with veil interlayering was observed. The localisation of the fibre failure, due to the inclusion of the veil, resulted in kink band formations which were found to be predictable in previous micro buckling models. The localisation of failure by the veil interlayer resulted in a pseudo-ductile response increasing the strain before failure by 24% compared with control samples.

## 1. Introduction

In 2019, the European Union mandated the reduction of at least 55% CO_2_ emissions by 2030 [1], and with the increasing weight of electric vehicle batteries [2], there is a greater need for lightweight structural materials in the automotive industry [3,4]. Further legislation exists in 14 countries to eliminate the sale of internal combustion engines by 2030, and the need for lower costing lightweight composites will increase within the decade [5]. Carbon fibre (CF) offers an alternative to traditional steel, as 60% to 80% weight reduction is achievable [6]. However, components made from composites are costly, as steel is only $7/kg, while CF costs between $25–50/kg [7]. The hybridisation of CF with lower costing Glass fibre (GF) offers a significant cost reduction, with GF costing only $3/kg while maintaining mechanical performance. Significant cost benefits of composite hybridisation for automotive structures, as the increased modulus to maintain part geometry from CF [8,9] and lower cost for GF lead to an overall more cost-effective component [6,10,11]. Issues with hybridisation include delamination at the fibre interface due to the difference in the elongation behaviour of the fibres [9]. Longitudinal delamination occurs, with several stacking sequences resulting in a negative hybrid effect compared to that rule of mixture [8,9]. 

Pseudo-ductility is the reorientation of fibres under strain to achieve a higher failure strain, resulting in nonlinear behaviour [12]. The veil is used at the interface of the fibre/matrix to stabilise the failure, as ultra-thin carbon fibre reinforced plastic (CFRP) layers have exhibited pseudo-ductile behaviour [13]. The hybridisation of three fibres aims to promote pseudo-ductile failure by using nanofibre layers to contain delamination and buckling that lead to premature failure [14].

Pseudo-ductility in hybrid composites is examined with a PA 6,6 veil at the hybrid interface. A PA 6,6 veil at the hybrid interface increases the fibre area without impacting the modulus of the hybrid composite. Previous authors found that veil interlaying reduced crack propagation during dynamic loading showing an improvement in stiffness degradation [15] and compression after impact behaviour [16] of composites, by mitigating the damage area. Rubber particulates are shown to reduce the storage modulus [17], which is detrimental to load bearing structures. The randomly oriented polyamide (PA 6,6) nanofibre veil interlayering is used to address the shortcomings of delamination through changing failure mechanics by improving Mode I+II and improving interfacial toughness between fibres. The flexural modulus of the hybrid composites is highly dependent on the stacking sequence of carbon fibre and glass fibre layering [11,18,19]. A significant negative hybrid effect for flexural strength occurs when carbon fibre is under compression due to premature buckling [11].

Compression failure caused by out of plane buckling is the most common type of premature failure in the interlayer hybridisation of carbon/glass fibres [8,9,14,20]. In unidirectional fibres, kink band formation after 99% structural loading tends to occur with a z to x direction tilt of the fibre direction [21]. Typically, the failure area of fibre kinking is localised compared with longitudinal cracking causing buckling [22]. The kink banding represents a gradual failure with the formation of the kink band being the final failure. Gradual failure creates nonlinear failure mechanics leading to higher strain making the composite pseudo-ductile [21]. Previously, nanofibres have been used in compression after impact applications reducing failure area, in carbon fibre composites [23,24,25,26,27] and glass fibre [28]. This research will aim to create a more stable failure with veil interlayering to promote more ductile composite, by localising the failure area. The addition of a veil nanofibre improves the fracture toughness of the CF/GF interface, to examine the effect on pseudo-ductility and further investigate the mechanism of failure [29].

Czel et al. [29] determined load bearing compression/tension plies need to have a higher strength than the applied loading after initial fibre/matrix failure. Additionally, the interlaminate fracture toughness (ILFT) at the interface is required to be stronger than the critical release energy. Increasing the ILFT through veil interlayering allows a thin fibrous layer to surround the failure area localizing crack propagation. Shekar et al. [30] showed that glass fibre reinforced plastic (GFRP) improves the Mode I ILFT behaviour with PA 6,6 nanofibre leading to a 68% increase in flexural strength. Tsotsis et al. [16] showed that PA 6 veil toughening in the interlaminar region, causes Mode II shear failure rather than Mode I crack opening. PA 6,6 veil by Beckermann et al. [31] showed an increase in CFRP fracture toughness for Mode I crack opening by 150% and Mode II by 50% with 4.5 g/m^2^ veils. The addition of nanofibres increases the energy required for crack propagation, as the matrix can tolerate higher stress concentrations. Mohammadi et al. [32] showed matrix cracking is reduced by 92% for CFRP, using electrospun PA 6,6 as an interlayer toughener. Blythe et al. [15] showed that PA 6,6 veil nanofibre localised crack propagation during flexural cyclic loading led to reduced stiffness degradation but was insufficient to prevent high loading amplitude catastrophic failure. 

There has been insufficient work characterising the carbon/glass fibre failure mechanism with increased ILFT. As the closest work by Dehrooyeh et al. [33] has investigated the nano mechanics of carbon nanotubes shown crack bridging with short nanofibres to be advantageous. This paper focuses on a comparative study of the bending failure mechanisms within toughened interlayer hybrid composites. Flexural modulus is optimised using stacking sequences of interlayered hybrid of carbon and glass fibre plies. The unique properties of this three-interlayer hybrid are investigated for a pseudo-ductile response, to make hybrid composites more suitable for automotive load bearing spring applications. The use of fibres to localise crack propagation will be the focus of the paper, as there has been minimal work examining failure area in bending in multi-fibre hybrids.

## 2. Materials and Methodology

### 2.1. Materials

The materials of this study include an intermediate modulus carbon fibre prepreg of 295 g/m^2^ areal density, with weft glass fibres of 5 g/m^2^. The E-Glass fibre prepreg of 845 g/m^2^, including woven ±60-degree 5 g/m^2^ glass fibres for handling. The glass fibre prepregs used the same 270 low temperature curing epoxy resin, sourced from GMS composites Australia. The polyamide 6,6 nanofibre veil is the Xantu.layr^®^ (Revolution Fibres, Aukland, New Zealand), which is randomly oriented at 4.5 g/m^2^ areal density. 

### 2.2. Sample Preparation

Using a guillotine, composites are cut to a 200 × 200 mm^2^ plates. The fibre mats were hand layered in a 200 × 200 mm^2^ aluminium frame, with a small gap between the edge of the bracket and the fibre to allow for resin flash and fibre spreading during compression moulding. The composites are moulded using an over stacked spacer to 2.4 mm, the material was then compressed to a height of 2 mm to ensure a proper lamination, as shown in Table 1. The compression resulted in an average stacking thickness of 1.98 mm. The samples were cured at 100 °C for 10 min and post cured over 10 min to 130 °C at a rate of 10 °C/min. The curing temperature was chosen from the manufacturer’s recommendation for isothermal curing. The control consisted of 6 plies of the 0.40 mm carbon fibre prepreg and 3 plies of the 0.80 mm glass fibre prepreg. The neat resin was cured using a 100 × 20 × 4 mm compression mould for the same cycle time, yielding flexural strength of 100–116 MPa compared with manufacturing data of 130–140 MPa. 

The prepreg was layered at room temperature with a Xantu.layr^®^ veil stacked between the glass and carbon interface. There was no measurable change in the stacking thickness, with the areal weight of the veil accounting for only 0.0025% of the total areal weight of the composite laminate. The curing temperature with and without the veil is examined in differential scanning calorimetry (DSC), with the results shown in Figure 1.

The DSC from the veil toughened sample show no additional change to the glass transition temperature with the addition of the nanofibre veil.

The meso-level analysis shows no observable variation in fibre alignment or nesting with different stacking sequences or with the veil inclusion. The veil is stacked between the glass-carbon interface rather than between carbon-carbon or glass-glass interfaces. The veil nanofiber is layered between prepreg stacks according to Table 2. 

### 2.3. Quasi-Static Testing of Composite Test

Quasi-static testing was carried out following ASTM D7264/D7264-21 [34] shown in Figure 2 for flexural testing at a rate of 1 mm/min using the Zwick UTM with a 10 kN load cell. 10 samples per stacking sequence were cut to 80 mm by 20 mm coupons using water jet cutting to avoid major fracturing. The geometry of the coupons fits a three-point bending setup and satisfies a span-to-thickness ratio of 1:20. The interlaminar fracture toughness samples are tested as per ASTM D5528-13 [35] at a rate of 5 mm/min using a 150 mm sample width with 2 mm depth and 50 mm defect made with Aluminium foil.

### 2.4. Failure Mechanism Analysis and Void Content

SEM imaging of the fractured surface and further in plane damage area was captured using the Zeiss Supra 40VP at 10 kV (Oberkochen, Germany), using sputter coating of 10 nm of gold on the samples. Surface morphology and in-plane damage area analysed using the BX61 Olympus microscope (Tokyo, Japan), with the software package stream motion version 2.5.2. Void content was measured using the 2D digital micrographs, 8-bit greyscale selection in open-source ImageJ™ software version 1.53 to calculate void content from binary images. 

Kink band formation and fibre buckling are failure mechanisms investigated under optical and SEM microscopy. The kink band is a formation caused under compressive strength where the fibres tilt from the X plane to the Z plane. Buckling is a common failure from axially loaded fibres, resulting in a chevron or V-notch occurring. In flexural testing, the fibres under compression delaminate fabric result in buckling. A V-notch is commonly associated with two instances of fibre failure resulting in a failure area. Additionally, elastic-micro buckling can occur, where the entire ply delaminates and micro buckles occur along the ply. Fibre crushing failure is another compression-based failure where the fibres do not tilt when failing and instead mushroom into each other.

## 3. Results and Discussion

### 3.1. Flexural Modulus and Rule of Mixture Calculations for Hybrid Toughened Composites

The rule of mixture (RoM) is used to determine the change in flexural modulus for the hybrid composites, compared with the theoretical values for 1:3 carbon to glass fibre. This will be used as a base line improvement for the flexural modulus with the addition of a veil. Equation (1) is used to calculate the theoretical flexural modulus of the material based on the RoM for upper and lower bounds methods.
(1)$$(A)\;E_{\mathrm{FRoM}}=V_{f}E_{f}+(1-\;f)\;E_{M}\;(B)\;E_{\mathrm{FRoM}}=\;{\left(\frac{f}{E_{F}}+\frac{1-f}{E_{M}}\right)}^{-1}\;$$

Equation (1) Theoretical flexural modulus of the material given the rule of mixture. (A) Upper and (B) lower bounds methods. 

Where E_FRoM_ is the elastic modulus calculated form the rule of mixtures V_f_ is the volume fraction of fibres E_f_ is the modulus of the fibres f is the hybrid fraction and E_M_ is the elastic modulus for the matrix.

Where f is the fibre volume fraction of each composite given by optical analysis, then derived in Equation (2), to give for the hybrid volume faction. The modulus predicted by the theoretical model predicts an upper bound of 42.37 GPa and a lower bound of 41.20 GPa for the fibre volume fraction used in this paper.
(2)$$r_{h}=\frac{h_{g}V_{\mathrm{fg}}}{h_{g}V_{\mathrm{fg}}{+h}_{c}V_{\mathrm{fc}}}\;$$

Equation (2) Hybrid volume fraction of CFRP and GFRP.

The variable are as follows r_h_ is the h_g_ and h_c_ is the stack hight for glass fibre and carbon fibre plies, respectively. V_fg_ and V_fc_ are the fibre volume fraction for glass fibre and carbon fibre, respectively [18].

For Flexural strength the calculations were as follows
(3)$$\sigma_{\mathrm{FRoM}}=\sigma_{\mathrm{Fc}}\;(1-r_{h})+\sigma_{\mathrm{Fg}}{\;r}_{h}.\;$$

Equation (3) Flexural strength a from rule of mixture.

With σ_Fc_, σ_Fg_ being maximum flexural strength for control CFRP and GFRP, respectively.
(4)$$\varepsilon_{\mathrm{FRoM}}=\varepsilon_{\mathrm{Fc}}\;(1-\;r_{h})+\varepsilon \text{\_}Fg_{\;}r_{h}$$

Equation (4) Flexural strain from rule of mixture.

$\varepsilon_{\mathrm{Fc}}$ is the flexural strain of CF control and $\varepsilon_{\mathrm{Fg}}$ is the flexural strain of GF control samples.

The effect of hybridisation is then calculated from the ratio of experimental values to Equation (3). The σ_FRoM_ was then calculated to be 900.72 MPa. The hybrid effect of the flexural strain was calculated from Equation (4) to be 0.01927 mm/mm.

### 3.2. The Veil Matrix Interactions and Effect on Curing

To investigate the residual stress hypothesis for negative hybrid effects the cure kinetics of CFRP and GFRP with a veil are investigated. Initial Mode-I testing of the veil, in Figure 3 of the curing with dwell times of ten minutes at 100 °C and 130 °C yields the highest fracture toughness for testing. At higher than 130 °C temperatures the resin did not flow through the veil resulting in a dry layer forming as the veil was not incorporated into the matrix, resulting in poor performance. As 150 °C resulted in a fracture toughness for G_IC_ of from 0.15 kJ/m^2^, with coupons cured dynamically from 100 °C to 130 °C, the fracture toughness increased to 0.22 kJ/m^2^. Mechanical testing aligned with the manufacturing data sheets for the prepregs supplied by the manufacturer. The data sheets supplied showed a fracture toughness of 0.15–0.2 kJ/m^2^ for curing ranges of 90 °C to 150 °C, indicating that the material is cured correctly. It has been found that temperature above 130 °C curing resulted in a 275% decrease in interlaminar fracture toughness. The void and dry spot content have been confirmed with SEM and microscopy, where the uncured samples show the veil as a clear separate layer. Under void analysis microscopy, the visual inspection of the fibre matrix layer indicates that there is no evidence of dry layer of the veil as no additional voids are detected around the matrix. The incorporation of the veil leads to Mode I increase of 61.81%, from 0.503 kJ/m^2^ to 0.832 kJ/m^2^_,_ for crack propagation after 30 mm of delamination. 

An examination of the surface using FTIR indicates the area under the artificial defect in Mode I tests show no signal of polyamide 6, 6 as the insert blocks the incorporation on one side of the matrix. The veil remains as a physical fibre and not integrating into the matrix, unlike other melt in veils. The area at the crack initiation found a strong signal at 3300 cm^−1^ assigned to the N-H stretching from polyamide 6,6. Aliphatic C-H peaks are common to both cured epoxy and PA 6,6, being observed in the FTIR spectrum with 2950 cm^−1^–2850 cm^−1^ split peaks.

The micrographs taken for void analysis concluded that the inclusion of the veil shows no significant change in void content in the interlaminar region with 3.69 ± 0.65% and 3.99 ± 0.42% control and veil toughened, respectively. The glass fibre is shown to have 3.785 ± 0.57% and with veil 4.1802 ± 0.56%. As both control and veil toughened carbon fibre fall within a deviation of the other, it is concluded that the void content remains unaffected by the veil.

### 3.3. Initial Failure Mechanism of Toughened CF GF Fibre

Three-point bending was used to analyse the effect of veil toughening on failure mechanisms of hybrid stacking sequences. As veil toughening increases the energy required for cracks to propagate, micrographs are used to determine the crack and delamination length at the catastrophic failure point. 

The yield strength of hybrid fibres is often lower than the ultimate strength of hybrid stacks due to the multilayer construction. The incorporation of veil toughening will examine how failure occurs at yield and ultimate strength. The veil reduces the matrix cracking and matrix/fibre debonding, in which the increased crack propagation energy leads to a reduced crack growth length [32]. The stacking sequences yield strength and strain at yield have been determined, with the control samples being glass and carbon fibre of the same stacking thickness. The final glass fibre stack has an aerial weight of 2550 gsm, compared to the carbon fibre’s aerial weight of 1800 gsm at 2.4 mm stacking height. Glass fibre shows a 6% difference in flexural strength, with carbon fibre being 29.4% lighter for the same thickness, as shown in Figure 4.

The flexural modulus of the control shown in Figure 6 GF is 37.56 ± 3.02 GPa and that of the control CF is 54.46 ± 3.97 GPa. The incorporation of veil toughening results in the flexural modulus of the control GFT increasing to 40.73 ± 3.72 GPa and the control CFT increasing to 58.86 ± 5.02 GPa. More significantly, Figure 4 shows veil toughened glass fibre has a 33.1% increase in strain at yield compared to control GF. The control GFT showed a decrease in flexural yield strength from an average of 888.27 MPa to 841.68 MPa for control GF and control GFT, respectively. 

Figure 5, shows a typical stress strain curve for CF and CFT, in which the flexural strength maximum is to be within the standard deviation, suggesting that veil has minimal impact on the flexural characteristics for carbon fibre. The GF sample shown in Figure 6, has a wider crack head than the CF, CFT and GFT suggesting a more gradual failure.

The increase in strain in Figure 6 shows a brittle failure rather than a gradual rack propagation. The brittle failure resulted in an increasing strain, compared with the gradual failure without veil.

The flexural modulus of carbon fibre undergoing compression, as shown in Figure 7, shows minimal reduction. With a single layer of veil interlayer in sample CCTGG an increase of 5.17% in flexural strength was recorded. However, there are significant improvements in the CTGTCTG structure in flexural modulus increase from 12.49 ± 1.13 GPa to 19.76 ± 1.67 GPa. 

For the CGCG structure, the flexural modulus increased by 19.69% and 7.69% for yield strain with a veil. No significant flexural strength change was recorded, with a 1.00% difference with toughening. For CGGC, there was a 7.4% reduction in flexural modulus; however, there was a strong overlap between the yield strength of the material, increasing from 645.57 ± 33.51 MPa to 657.85 ± 30.92 MPa. The yield strength was 18.8% lower than the ultimate yield strength, and there was delamination in the compressive carbon fibre ply with the central glass. The failure mechanism remained unchanged with veil toughening. 

When carbon fibre undergoes compression in Figure 8 there is minor failure under 0.02 mm/mm in CGGC and CTGGTC there is non-linear failure after.

It was expected this would cause catastrophic fibre failure; however, the glass carbon layer on the tension side was able to support loading, as shown in Figure 8. The structure CGGC experienced a 0.013 mm/mm deflection, after which the top CF and GF plies prematurely failed. Beyond the yield point, the CGGC structure increased in flexural strength as the glass and carbon fibre on the tension side could maintain loading. Maximum flexural strength occurred at 0.024 mm/mm, where the glass and carbon fibre experienced combined failure.

The pseudo-ductile response ends when the glass fibre ply forms a kink band. The stress strain curve of the CTGGTC suggests that gradual matrix cracking occurred until the kink band formed, greatly reducing load bearing. This gradual formation is pseudo-ductile as the linear section of the stress strain curve changes after minor matrix cracking, leading to a failure strain exceeding 0.03 mm/mm and initial failure yielding at 0.013 mm/mm. 

Figure 9 shows the strain curve when carbon and glass fibres are dispersed by interlayering of fibres CGCG and CTGTCTG. The result of dispersion is a less defined failure peak as the lower yield strength results in a layer-by-layer failure.

As the carbon and glass fibre curves partially overlap there is a distinct ply by ply failure in the CGCG and CTGTCTGTC laminates. The flexural loading in Figure 9 is low enough for the failure laminate to continue loading as the next layer undergoes compression failure.

The stress–strain behaviour shown in Figure 10, of dispersed plies, compared with complete separation as in CCGG and CCTGG shows 2 defined failure stages. Starting with carbon fibre compression failure, however, the glass fibre under tension undergoes a secondary failure leading to the secondary nonlinear curve.

However, the fibre failure at the higher yield resulted in a drop in material property, leading to a lower maximum flexural strength. The maximum flexural strength of the veil toughened material, shown in Figure 10, was experienced at 0.05 mm/mm. The glass fibre experienced tension rupturing, causing the final failure mechanism and a flexural strength drop.

Glass fibre with higher elongation in the compression layer results in hybrids with higher flexural strength than lower elongation carbon fibre. The flexural strength of the glass fibre in the compression layer ranges from 600–900 MPa compared with the 400–700 MPa of carbon, as shown in Figure 11. 

The veil toughening had a positive yield strain and negative flexural modulus effect on the GCGC. The flexural modulus of the GCGC was 11.23% lower for veil toughened samples. However, the strain at yield for the GTCTGTC sample increased from 0.018 ± 0.00014 mm/mm to 0.025 ± 0.00014 mm/mm, a 24% increase. When comparing the flexural modulus of GGCC with GGTCC, the GGTCC with veil toughening showed an increase of 6.6% on average. More significantly, GGTCC flexural strength was increased by 15.81%. The strain increases of GFT and CTGCTG are attributed to the veil within the resin preventing crack propagation.In comparison to the carbon fibres under compression glass fibres under compression have a wide yield failure peak. Figure 12 shows a single elongated peak, suggesting that all layers fail after failure initiates.

**Figure 12 materials-15-08877-f012:**
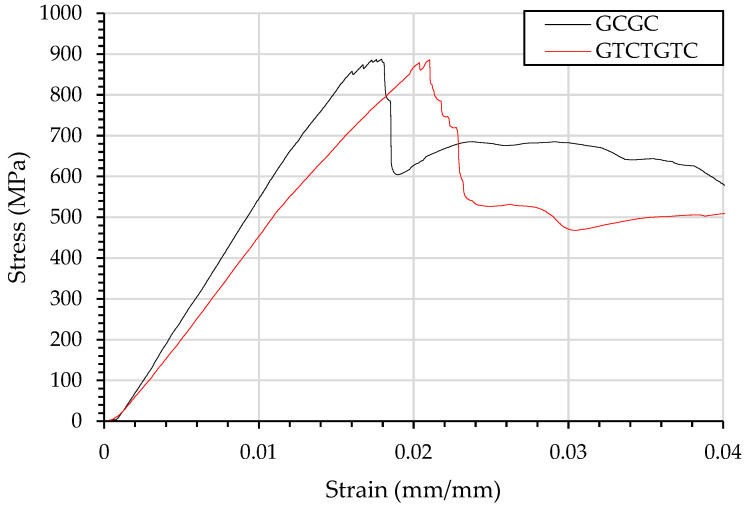
Flexural testing using three-point bending of composites beams and resulting stress strain curve of GCGC and GTCTGTC.

Both alternating samples have flexural strain exceeding the carbon fibre yield strain of 0.0123 mm/mm, with 0.018 mm/mm and 0.025 mm/mm for GCGC and GTCTGTC, respectively. As the flexural strain increases, the extension of the glass/carbon interface results in tension delamination.

The increased Mode-I showed the highest improvement to the strain multiple interfacial hybrids, where the difference in strain characteristics was most likely to cause delamination. In single interface samples, the veil acts to reduce the delamination growth; this causes a higher yield strength as the delamination growth is minimised to the point that the fibres will then fail.

The stress–strain curves of GGCC and GGTCC have much narrower failure peak shown in Figure 13. Compared with carbon fibre undergoing compression, glass fibre can bare loading at greater stress and strain. 

Figure 13 suggests that the dispersal of fibres is less effective than the stacking sequence for flexural loading, with the veil aiding in yield strain. The flexural strength of the composites with the veil was shown to remain the same as the control samples. However, the strain of the composites is shown to improve.

To characterise the flexural characteristics of the composites, the RoM is used to investigate the effect of veil toughening. When the flexural modulus is subjected compared to the rule of mixture for alternating samples, an increase in yield strain was observed, which also showed a negative modulus hybrid effect. The hybrid effect of the flexural modulus was 7.16%, 5.13%, and −229.8% for CCGG, CGGC, and CGCG, respectively. Compared with toughened hybrid effects of 8.796%, −1.92%, and −108.4% for CCTGG, CGGC, and CTGTCTG, respectively. The stacking sequence which showed the highest improvement was CGCG, with less reduction in flexural modulus. The sandwich structure CGGC showed a change in failure mechanism with veil toughening but no strength change, which indicates that veil toughening, on the tensile side has less impact on the overall properties. Samples that have glass subjected to compression showed similar trends with increased yield strain with the interlayering of the veil. The energy increase resulting from the crack bridging of the veil translates to an increased yield strain. For GCGC and GTCTGTC, a negative hybrid effect was observed for flexural moduli of −10.24% and −24.21%, respectively. However, GGCC and GGTCC had a positive hybrid effect for all flexural characteristics, with hybrid effect for the flexural modulus of 1.18% and 2.75%, respectively.

Figure 14 shows the rule of mixtures estimation of the hybrid stacking sequences. The RoM shows poor correlation with the strain of the hybrids, as carbon fibre undergoing compression tends to behave with the characteristics of the CF sample. The sample with the least correlation to RoM is CGCG, where the dispersion of fibres with carbon fibre undergoing compression leads to poor modulus and strain.

As shown in Figure 14, GGCC and GGTCC are the idealised versions of hybridised CF: GF for the RoM. The RoM is stacking-dependent for glass and carbon fibre’s flexural strength and modulus. Increasing the interfacial toughness improved the strain to failure of carbon fibre undergoing compression samples. The carbon glass interface has a change in material stiffness between fibre matrix, causing an uneven stress distribution.

The veil toughened samples showed a higher correlation to the RoM than the untoughened samples. This correlation would suggest that the negative hybrid effect is based on fibre prematurely failing, which improves predictability with RoM.

The highest achieved flexural strength occurred when the glass fibre was subjected to compression stacking sequences. These fibre stacks showed good correlation to the previous hybridisation stacking sequences and followed the optimal fibre stacking sequence [9,11,36]. Veil toughening showed the most mechanical improvement in alternating structures with 3 carbon/glass fibre interfaces. When compared to previous hybrid trends, hybrid interfacial toughness is more impactful than increasing the compressibility of the ply when subjected to compression for flexural strain and flexural strength. Improving matrix toughness showed the highest impact with dispersed interlayers. However, following the hybridisation optimisation, it yielded the highest improvement, as delamination resulting from fibre shearing between interfaces is minimised with veil toughening.

To characterise why veil toughening changes material characteristic microscopy is employed to investigate failure mechanisms. Key to understanding why the previously employed modes overestimate is the veil toughening changing the failure mechanism. The next section will evaluate on the failure mechanisms and how they will change with increased interlaminar fracture toughness. 

### 3.4. Fracture Mechanisms within Interlayer Toughening

The change in failure mechanisms due to the introduction of the veil is analysed using SEM. As veil toughening deflects the crack propagation, the effect of the veil toughening on the matrix is examined for the change of failure propagation and buckling delamination within the hybrid samples. Figure 15 shows the investigation of the failure mechanism using micrographs of the samples to determine the crack propagation.

The control CF sample in Figure 15A, started with delamination propagating to cause buckling of the fibres initiating from resin rich zones. Figure 15B, the CFT sample shows that there is no buckling, instead of which, failing is by fibre kinking. The failure mechanism of the control CFT showed no yield strain increase, when compared with the control CF. The control CFT samples showed there is more extensive delamination between fibre layers and propagation that occurred with fibre kinking. 

The resulting change to the failure mechanism with minimal change in characteristics is different to the glass fibre composites, where both have crushing failure shown in Figure 16.

Figure 16B shows that the change in failure mechanism changed the mechanical behaviour of the glass fibres in the stress–strain curve. GF showed a progressive crushing failure of each layer under compression force, resulting in elongated cracking until tension rupturing resulting in a drop to <800 MPa force. The veil toughened sample failed as one peak; failure was then reached at the same post-failure condition as GF after 0.03 mm/mm strain. The GFT sample showed higher retention of loading than did GF after failure.

The control GF and control CF samples fail by localised compression layer fibre failure causing catastrophic failure. In veil toughened carbon fibre samples, the failure was due to fibre kinking leading to flexural yield. Glass fibre was shown to have a larger tension rupturing area as failure occurred at a higher strain. Figure 16 B. As each ply of GF has a high thickness, the material’s capacity to maintain loading after yield is relied on to produce pseudo-ductile behaviour.

The control GF failed due to crushing failure in a localised area. The control GFT has a change in yield strain with an increase of 25.08%. It was observed that there was a greater area of crushing and tension rupturing within the control GFT.

In hybrid composites, the compression failure of the top ply led to a stress concentration on the next layer, causing delamination to propagate as the fibres failed. Despite no significant change in flexural strength, there was a distinct failure mechanism change within the hybrid structures. The inclusion of veil interlayers between the carbon glass interface resulted in the delamination growth, which required a higher stress concentration, typically leading to an increase in strain before yielding.

Figure 17A show CGGC with tension failure of the glass fibre layer. This failure separates the top carbon fibre ply from the glass fibre ply. The V-notched fibre buckling occurs through both the carbon and glass fibre, causing gradual failure. 

In Figure 17B, the CTGGTC sample shows a change in failure mechanism with the inclusion of veil instances of buckling that changed to fibre kinking. Veil toughened composites reduced the out-of-plane buckling caused by compression, resulting in shear kinking being the dominant failure mechanism. In veil toughened samples, there is a lower flexural strength; however, past the yield point, there is a pseudo-ductile response.

Figure 18A,B are more pseudo-ductile in stress–strain with the gradual failure mechanism shown as elastic and V-notch buckling. However, both stacking sequences show significate negative hybrid effects for all characteristics. From the micrographs in Figure 18A,B, the damage sustained by the carbon layer is irreversible at yield.

Figure 18A,B both show layer-by-layer failure, with CGCG featuring interlaminar failure and CTGTCTG crack propagation, though of the fibre rather than the matrix. When undergoing compression, the carbon fibre failed from delamination of the top layer, causing crack development through the glass to the second carbon fibre layer. Tension failure of the CGCG sample was observed as the elastic buckling resulted in flexural failure to the second carbon ply. The cracks propagated around the cross stitching towards the second carbon later, resulting in ply splitting and tension side delamination. On average for CGCG, elastic buckling resulted in a sharp drop in flexural strength, only increasing at higher strain when the final glass fibre ply began tension loading. Compared to the toughened CTGTCTG, the inter-fibre failure resulted in less fibre failure and more matrix fibre debonding. 

As the veil enhances the matrix toughness, the CTGTCTG in CTGTCTG in Figure 18B shows a significant failure on a micro-level between fibres rather than within the matrix. The failure mechanism indicates that the crack propagation has been deflected from the matrix into the fibres. The delamination on Figure 18B shows that the veil traps crack propagation as marked D, this shear failure shows matrix fibre debonding at the interface. The tension side delamination was observed with both CGCG and CTGTCTG where at higher strain the glass fibre over elongates.

Figure 19A shows a similar elasic buckling failure mechanism, which is common to the carbon fibre undergoing compression.

Figure 19A CCGG depicts elastic buckling, this was induced when the carbon fibre undergoes compression in the presence of crimping fabric. The crimping of the fabric in the E zone, indicates waviness of the fabric induces buckling. The carbon fibres are shown to undergo compression failure in the alternating sample, where the elongation of the crack head resulted in the failure overlapping with the glass undergoing tension. The overlap of the glass and carbon fibre peaks then resulted in pseudo-ductile behaviour, whereby the material behaves nonlinearly. Additionally, shear failure of the second carbon fibre ply resulted in additional kink band formation and out-of-plane deformation, resulting in extensive fibre failure offset from the centre.

Figure 19B shows CCTGG with two areas where crack initiation is observed, which was caused by kink band formation rather than elastic buckling. The elastic buckling was caused by extensive delamination followed by compression failure. CCTGG showed no elastic buckling but the CC interface was untoughened, thus the CG interface was the source of buckling. Instead, the fibres form a kink band, causing a drop in loading. Much like the previous CTGTCTG, the extensive delamination resulted in a higher strain before yield. 

With carbon under compression, buckling is the primary yield failure. The addition of veil interlayering changes the failure mechanism of the compression carbon fibre layer to shear-based kinking or inter-fibre debonding. CCGG, CGGC, and CGCG have a large delamination area caused by the carbon fibre ply buckling due to crack growth. In the compression layer, carbon fibre tends to buckle without veil interlayering. After yield failure, the dynamic stress concentration on the local region leads to a catastrophic failure. As the carbon fibre has the highest flexural yield, the glass fibre immediately fails, leading to tension delamination between the glass/carbon fibre interface. Fibre failure in the veil toughened CCTGG, CTGGTC, and CTGTCTG was shown to deflect the crack propagation path at the fibre interface point. For CCTGG, propagation was through the carbon/glass fibre interface. The alternating CTGTCTG sample had cracks that continued the propagation through the glass fibre tow rather than the interlaminar region.

This behaviour is similar to the pseudo-ductile effect, with low quantities of carbon fibre surrounded by a more ductile fibre. In this case, the veil acted as the ductile fibre, and the fibre breakage of the carbon fibre was contained by the veil as the buckling failure was significantly smaller in the failure area, allowing the composite to behave in a pseudo-ductile manner. The behaviour exhibited by the CGGC composites is unlike the fibre fragmenting that is typical of the pseudo-ductile composites [37]. Rather, at the yield point, the yield strain of the overall composite is increased due to pseudo-ductility. 

In previous research, glass fibre undergoing compression has shown a higher flexural loading [9]. As shown in Figure 19 the failure tends to be within the matrix, compared with Figure 20 where the buckling is a result of delamination, with the carbon fibre undergoing compression failure first. Samples with carbon fibre undergoing compression show crack initiation, which is caused by low flexural strain with glass fibre over elongating to cause matrix failure, resulting in buckling of the carbon fibre.

The GCGC sample shown in Figure 20A had delamination between the glass/carbon fibre interface. Both the compression and tension side show failure when the glass fibre compression ply failed. The result of glass fibre failure was a transfer of stress concentration to the carbon fibre layer. The carbon fibre sandwiching glass fibre below the compression layer results in a similar flexural strength to that of the compression layer. Similar compression properties in carbon and glass fibre were observed, with the yield failure resulting in simultaneous catastrophic failure for alternating samples.

In the GTCTGTC sample, in Figure 20B, e carbon fibre ply remained intact with little tensile rupturing. Instead, the glass fibre failure resulted in delamination on the tension side. The buckling failure resulted in a localised failure through the centre of the laminate. The veil deflects cracks as they propagate, acting as a tertiary fibre layer and trapping crack propagation between veil and fibres, as shown in Figure 20C. When carbon fibre was placed off the compression layer in GTCTGTC, the crack propagation deflection from the veil and glass layers created a high strain before yield behaviour from the composite compared to GCGC. The main failure mechanisms causing an increase in strain was the change in failure from elastic buckling to a more localised buckling on the glass ply, with minor delamination between carbon and glass on the second ply, likely induced by fibre crimping. 

The dispersion of carbon and glass fibres leads to an elongation of the crack head. The strain with the veil is higher than without as longitudinal cracking is reduced due to crack localisation. Compared with the dispersed hybrids glass fibres under compression, Figure 21 shows the veil sample has no premature failure increasing the flexural strength. The sharper crack head of the veil samples indicates a complete failure. 

The variation in between GGCC in A and GGTCC in B shows a change in failure mechanism, from glass fibre buckling to fibre crushing leading to high delamination length between glass fibres. The crack propagation in Figure 21 for GGCC resulted in a >500 μm delamination length. In GGTCC, there was a similar delamination length to that of GGCC; however, the flexural strength of GGTCC is increased. Indicating that the matrix fracture toughness also increases the maximum failure strength when delamination is the primary failure mechanism. 

Figure 21B shows GGTCC fibres remained intact on the second glass fibre ply, with some waviness of the carbon fibre layers. The compression layer of glass fibre remained intact during crack propagation, with delamination from the carbon fibre ply due to crack propagation and shear. An increase in yield strain was observed for GGTCC; however, delamination length remains the same for GGCC, with its failure mode changing to fibre kinking. The delamination length remained the same, indicating that propagation is slower due to increased fracture toughness. Compared to previous hybrid studies the GGCC is shown to be the most optimal stacking sequence for achieving the highest flexural strength [9,14,36].

### 3.5. Fracture Mechanics Leading to Positive Hybrid Effects

Under optical microscopy of carbon fibre under compression with veil toughening, kink band formation was observed. Kink banding caused sudden failure rather than a progressive failure of the CF samples that buckled. Kink banding was present when the delamination was localised by the veil toughening. Typically, in the stress–strain curve, the kink band is shown as a non-linear section followed by a sudden drop as failure occurs.

Buckling causing localised fibre rupturing was observed with carbon fibre under compression. With glass fibres, the higher elongation under compression showed crushing failure. The veil toughening resulted in a gradual crush tension failure with buckling being reduced as longitudinal delamination is localised. With excess waviness, elastic buckling occurred and caused an elongated crack peak as the fibre failure is gradual, with a significantly wider damage area due to excessive crack propagation.

Changes to the Mode-I characteristics have resulted in the change in failure in hybrid composites. The failure mechanism of buckling was mitigated, resulting in a more gradual kink band failure. Longitudinal crack propagation is shown to cause the buckling as the fibres are pushed out of the plane under compression. Buckling of CF plies caused by fibre-matrix debonding was localised by veil resulting in more gradual kink band failure. The randomly oriented nanofibre veils prevent crack propagation, as shown in Figure 22, leading to reduced delamination. The localisation of failure stabilised the initiation of cracking resulting in higher strain before failure with interfacial toughening.

The crack propagation is redirected into the matrix by the interlayering of a veil, creating a longer crack pathway. The veil redirects crack propagation, leading to higher elongation and gradual failure in hybrids promoting pseudo-ductile responses. This redirection of preparation reduces the delamination extent of localising failure. The local failure results in a Kink band failure rather than buckling, as the matrix can hold fibres in place preventing buckling. 

When fibre failure occurs in veil toughened samples are at higher strain, thus, the resulting dynamic response of the next layer tends to result in catastrophic failure. When glass fibre is on the tension side and fibre dispersion is high, the result is a pseudo-ductile response after fibre failure, as the elongation is less than the glass fibre’s critical elongation. 

The failure of the CTGTCTG sample shows that the veil acts to localise failure as the crack propagation and was blunted by the veil. The result is crack propagation that was redirected through the matrix and interply. Localisation of failure is beneficial to composites in fatigue applications as non-visible damage is a critical concern. Localisation of matrix cracking leads to higher strain as the matrix crack can take more loading before failure, which is reached at a lower stress level without a veil.

The localisation of crack propagation results in stability of failure, thus the toughened samples have a higher strain before failure. In glass fibre stability of failure is most prevalent in GFT as the GF samples suddenly drop in flexural loading after yield, whereas GFT gradually decreases. 

Compared with other nanomaterials such as carbon nanotubes [33] the veil acts to blunt crack propagation as a layer. The continuous layer of veil prevents propagation from traveling across the fibre interface. Additionally, the pseudo-ductile behaviour is experienced in sandwiching panels, as other authors have determined [38]. failure of the compression layer with a thicker sandwich layer material doesn’t lead to catastrophic failure as the tensile layer is able to take loading.

To further investigate the positive hybrid effect CFRP is selectively layered with veil nanofibre.

### 3.6. CFRP with Selective Veil Layering’s

As the failure mechanism in hybrid fabrics lead to a change in mechanical behaviour, this section aims to investigate the layering of the veil and if the CFRP composite changes failure mechanics the same as the hybrid counterpart. For this, the CFRP is layered with the veil shown in Table 3. This aimed to replicate the behaviour of hybrid fibres using interlayer toughening instead of more ductile fibres by improving the compression strength of the material. To investigate the compression layers buckling as support for the change in failure mechanism hypothesis leading to positive hybrid effect. By using CFRP and the veils additional failure changes due to residual stress between fibre types are eliminated.

To demonstrate hybrid fibre stacking rules outlined previously by Wu et al. [8], the next section will selectively improve the layers responsible for maximum stress distribution in hybrid fibres, this being the compression zone and off-neutral axis. Using only CFRP and veil this section will investigate whether the veil acts using the same rules as the hybridisation with glass fibre.

A single-veil reinforcement in the compression zone has resulted in an increase to the compression strength of the material as it undergoes compressive strain changes the damage mechanism. In comparison to single-veil reinforcement, in the tensile zone, the difference in failure mechanism is visible. As the veil is placed on the interlaminar region, the force acting on the carbon fibre area does not have veil reinforcement. Where the veil reinforcement incorporated in the compression zone, added ductility within the matrix causes a change in the failure mechanism. With the veil reinforcement, minimal buckling is observed, whereas, with veil reinforcement of the tensile reinforcement, there is no toughening within the compression layer, which results in a higher degree of buckling. 

There is evidence in Table 4 that carbon fibre toughening increases the modulus; however, again in shear positions the CFRP flexural strength and modulus is reduced. Compared with the two failure modes of the control sample—where the material either fails under the machine head in layer-by-layer failure like the other toughened samples or fails early by crushing failure under the glass fibre cross stitching—the crack then propagates, causing chevron base failure. There are two separate failure mechanisms, resulting in the control A sample having a high variation in material property. However, the control sample set are separated according to their failure mode, resulting material characteristics have an improved reproducibility, suggesting stitching location causing premature failure. The flexural strength of the carbon fibre data sets statistically remain the same as the variation between veil toughening and the control is insignificant, as shown in Table 4.

Figure 23 shows that control sample A displays buckling type failure caused by Mode-I crack opening failure, determined by micrographs of the cross section. The crack propagates from the chevron-based Mode-I kink band failure, which then propagates into buckling of the material. The difference can be seen in B, where the toughening is on the top layer of the carbon fibre; therefore, the crack propagation is shear based. This shear-based failure results in carbon fibre shear failure compared to the V-notch failure of the untoughened sample in both sample A and sample B2. The B1 samples failed by a kink band formation, localised down the centre of the structure. 

Longitudinal cracking is noticeable on each layer of the sample causing the V-notch buckling of the top layer. B1 kink banding is longitudinal cracking on the neutral axis as failure propagation is dispersed by kink banding causing a shear deformation. B2 longitudinal cracking is reduced on the tension side wear veil into layering prevents further cracking. Sample C has a kink band formation and similar longitudinal cracking to the B1 sample where most of the longitudinal cracks appear on the tension side. In comparison to the veil in the neutral axis, cracks are redirected, and no V-notch buckling occurs, instead tension inside longitudinal cracking is present. In sample E, fibre failure was kink band causing delamination between plies.

Several samples have large areas of damage in a circular pattern which then propagates into a chevron-based defect which is visible around the glass fibre. As observed in Figure 23, the buckling occurs immediately due to the presence of stitching, which initiates the first section of the chevron-based kink band failure, as proposed by Wang et al. [22]. With the incorporation of the veil on the compression layer, there is a clear change in the failure mechanism towards the fibre tilting of kink bending. 

The failure mechanisms suggest that the shear behaviour of the veil thoughted samples is poorer than the compression behaviour as there is more delamination due to fibre failure on the neutral axis shown in Figure 23. The D samples was specifically created to investigate the shear behaviour. As shown in Figure 24 the D sample yields before the other samples.

From the stress–strain curve of the veil toughened carbon fibre, the control sample of carbon fibre fails via sudden buckling of the material in a chevron mode of kink band failure. The veil toughened samples fail ply-by-ply, both B2 and D have extended shoulders beyond the initial crack. Figure 24 shows veil toughening on the compression side leads to an increased modulus, in comparison to the tension side. The veil nanofibre elongates the crack head due to the localisation of failure while decreasing the flexural strength as the epoxy was increased in ductility. Multiple veils on the compression side show a premature failure leading to a lower average flexural strength. The neutral axis veil reinforcement leads to shear failure, as the reduction of short beam shear strength is recorded with a veil. This shear failure changed with the glass fibre introduction, where the increase in flexural strength is found with reinforcement on the compression side’s neutral axis. 

Carbon fibre with B1 layering shows a higher average than the A control, with no buckling present and the compression layer interlayering reduces the impact of early compression yield.

The GTCTGTC sample’s negative hybrid behaviour is explained by these carbon fibre tests whereby the veil reduces the shear strength of the neutral axis interlaminar regions. The shear failure leading to the negative hybrid effects supports the change of failure mechanism as a cause of negative hybrid effects. Given the reduction in shear strength, the next section will investigate the predictability of the veil toughened samples given the increase in modulus but the decrease in shear. 

### 3.7. Comparison of the Flexural Strength to Previous Kink Band Failure Models

The failure mechanisms of the composites have been previously modelled to predict the flexural strength by a derivation of the longitudinal compressive strength [18,39]. The experimental results for many fibre volume fractions for these are shown as an overestimate, as the assumption of the failure mechanism is either micro buckling or fibre kinking. To better estimate the hybrids failure the failure mechanisms are 

The Lo-Chim model uses compression strength based on the localised micro-buckling [40], as a majority of the buckling is non localised due to waviness it is expected that this model will be less accurate. The Lo-Chim model is shown below
(5)$$X_{C}\;=\frac{G_{12}}{1.5+12{\left(6/\pi \right)}^{2}\left(G_{11}/E_{11}\right)}$$

Equation (5) Lo-Chim model for micro-buckling Where X_C_ is compressive strength, G12 is the shear modulus and E_11_ is the longitudinal tensile modulus, respectively.

The Lo-Chim and Budainsky models were modified using G_IC_ characteristics to replace the shear values for G_12._ Previous work has shown the G_IIC_ modulus to be lower than the G_12_ which is being used to better estimate the failure.

The Budiansky model [41,42] for compression strength is calculated from the following for fibre kinking.
(6)$$X_{C}\;=\frac{G_{m}}{\left(1-V_{f}\right)\left(1+\phi /\gamma_{Y}\right)}$$

Equation (6) Budainsky model for fibre kinking modes. Where G_M_ is the matrix modulus, Φ is the misalignment angle and γ_Y_ longitudinal shear strain.

The flexible strength using the RoM was derived using Lo-Chim and Budiansky models and Figure 25 is the plot of hybrid volume fraction of glass fibre compared with carbon fibre. For this experimental the hybrid volume fraction is set at 0.3.

Lo-Chim showed an overestimation of the flexural strengths and was elaborated on previously in the literature [18] for hybrid fibres. The overestimation in the Lo-Chim model is with the carbon fibre, where the waviness of cross-stitching creates a buckling failure rather than the Lo-Chims localised buckling [40]. However, with a comparison to the hybrid Lo-Chim using the RoM characteristics for the calculation the GGTCC sample has a similar flexural strength. The correlation between GGTCC and Lo-Chim is due to the localisation of crack failure and buckling rather than the elastic buckling observed in GCGC samples.

In the case of the Budiansky model overestimates are on the hybrid. The lower angle of alignment and higher strain produces an overestimate. However, the Budiansky model uses shear strain the addition of a veil causes an over-prediction for CF and under predicted CFT while both had the same flexural strength but different strains. The yield strength of the hybrids came closer to the derived values for compressive strength rather than flexural. Ultimate flexural strength calculated from the predictive modelling of Lo-Chim correlated with only GGTCC. With Lo-Chim model for CF being closer to the compression strength than flexural, this would suggest the fibre shear and tension modulus are more accurate at predicting flexural strength in composites with higher waviness, suggesting that the yield is premature compression driven.

It would then further suggest that a mixed mode model may be more applicable to hybrids as the failure mechanism of carbon and glass fibre differs based on stacking sequence. 

## 4. Conclusions

The study of veil toughening and pseudo-ductility concludes that with veil toughening, crack localisation results in pseudo-ductile failure. Pseudo-ductility is observed in carbon fibres sandwiching glass fibres and carbon and glass fibres alternating, where the elongation of the crack head reaches a more stable loading condition. When carbon fibre undergoes compression with the veil interlayering, buckling was shown to change to a failure mechanism to fibre kinking. The veil interlayers within carbon fibre and alternating samples showed an increase in modulus of 8.07% and 19.69%, respectively, compared to untoughened samples. The increase in flexural modulus for single interfacial hybrids indicates that the veil toughening acted to reduce strain within the interlaminar regions. The veil acts as a randomly oriented fibre layer, as toughened samples were observed to have cracks propagating through interlaminar regions. The change in failure mechanism results in higher strain before failure, this supports the previously established hypothesis of failure development for hybrid fibre failure modes. 

The carbon fibre plies undergo compression, resulting in the failure of carbon fibre prematurely and elongation of the crack tip, further overlapping with glass fibre strain failure. Stacking sequences with glass under compression showed no pseudo-ductility, with the strain being increased linearly with stress. The failure mechanism change with increased strain suggests that premature failure, caused by buckling has been mitigated to produce a more predictable RoM hybrid. The veil toughening validates the consensus that buckling is a premature failure of the hybrid structure. 

Based on the results of this study, the veil toughening is shown to improve strain with highly distributed fibres. This study shows that, under compression, toughened high-elongation glass fibres can redirect delamination caused by over-elongation between interfaces. As a result, there was an increase of 15.81% in flexural strength, with the addition of a single veil between the carbon/glass fibre interface. This redirection results in improved flexural strength and strain by changing the propagation of the failure. 

Micro buckling is an expected failure mechanism, whereas the failure observed under optical microscopy was closer to delamination causing longitudinal cracking leading to buckling rather than micro-buckling. The localisation of the fibre failure, due to the inclusion of the veil, resulted in kink band formations which were found to be predictable in previous micro buckling models. As the failure of these hybrids is a compression-based failure the current models overestimate results as micro buckling leading to kink band failure is the ultimate failure. While shear failure is observed with veil interlayering on the neutral axis fibres leading to reduced performance.

When comparing predicted flexural strength using the GIIC characteristics, the Lo-Chim model is more accurate for compression failure. Strain was identified as an issue using the RoM for characterising the flexural strength. The flexural yield strain increased due to the veil localisation of the crack propagation resulting in a kink band, making more ductile and predictable composites.

## Figures and Tables

**Figure 1 materials-15-08877-f001:**
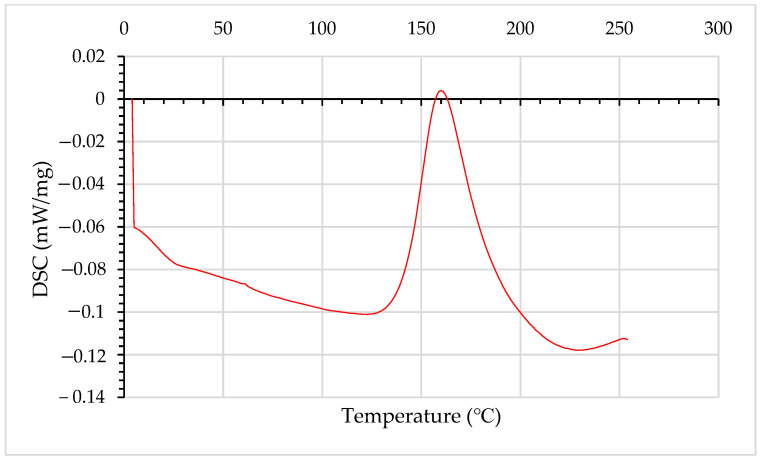
Differential scanning calorimetry, showing a peak at 155 °C consistent with manufacturing information.

**Figure 2 materials-15-08877-f002:**
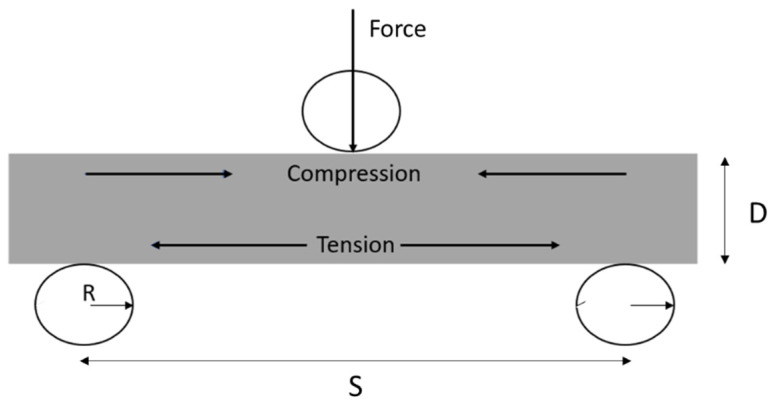
Schematic of three-point bending setup with radius (R) = 5 mm and Span of the grippers (S) = 40 mm to satisfy the ASTM span to thickness ratio of 1:20 and the depth of the sample (D) = 2 mm.

**Figure 3 materials-15-08877-f003:**
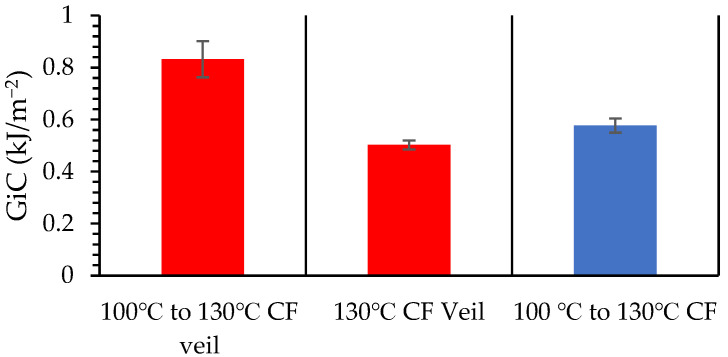
Interlaminar fracture toughness Mode I testing for carbon fibre composites isothermally cured at 100 °C and 130 °C, veil toughened carbon fibre isothermally cured at 130 °C, veil toughened carbon fibre isothermally cured 100 °C and 130 °C.

**Figure 4 materials-15-08877-f004:**
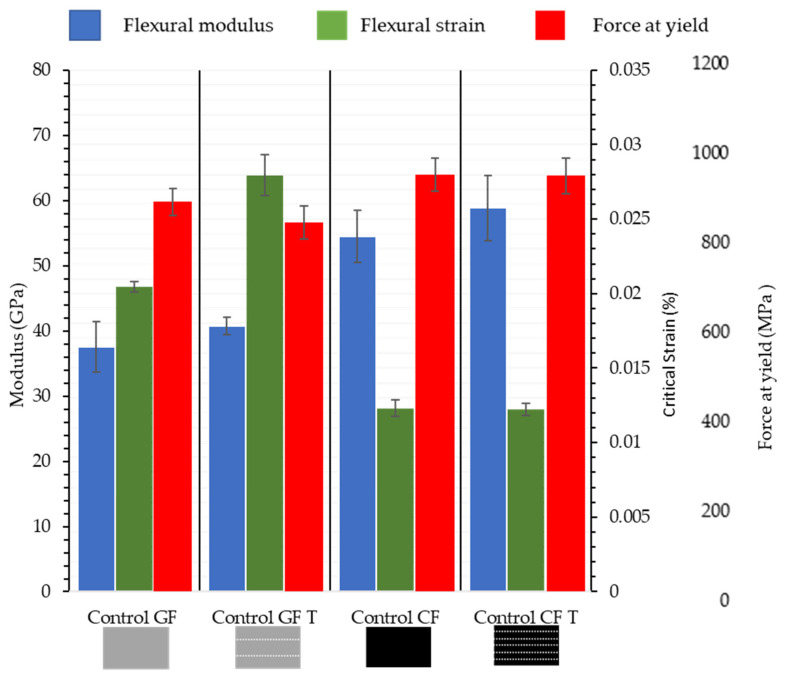
Yield strength of control samples of control Glass Fibre (GF) and Glass Fibre Toughened (GFT) and control Carbon Fibre (CF) and Control Carbon Toughened (CFT) fibre after three-point bending. Showing a strain increase with veil toughening for glass fibre. T denotes the interlayer veil toughening, where every layer of the controls used veil toughening.

**Figure 5 materials-15-08877-f005:**
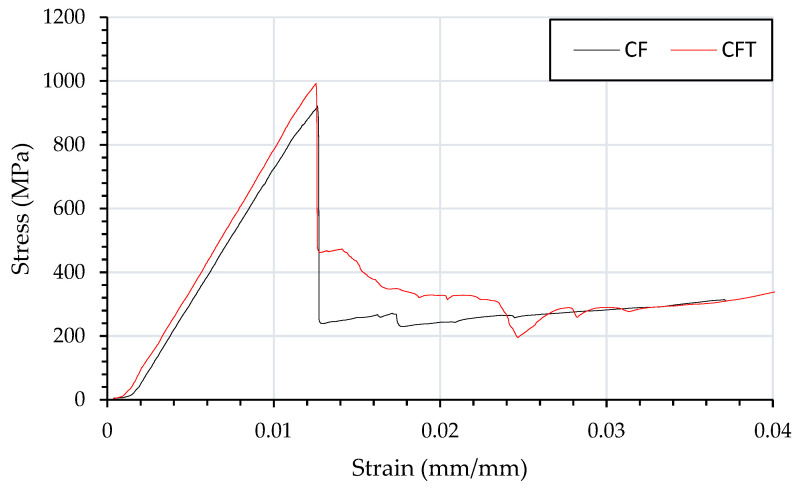
Flexural testing using three-point bending of composites beams and resulting stress strain curve of CFRP (CF) and veil toughened CFRP (CFT).

**Figure 6 materials-15-08877-f006:**
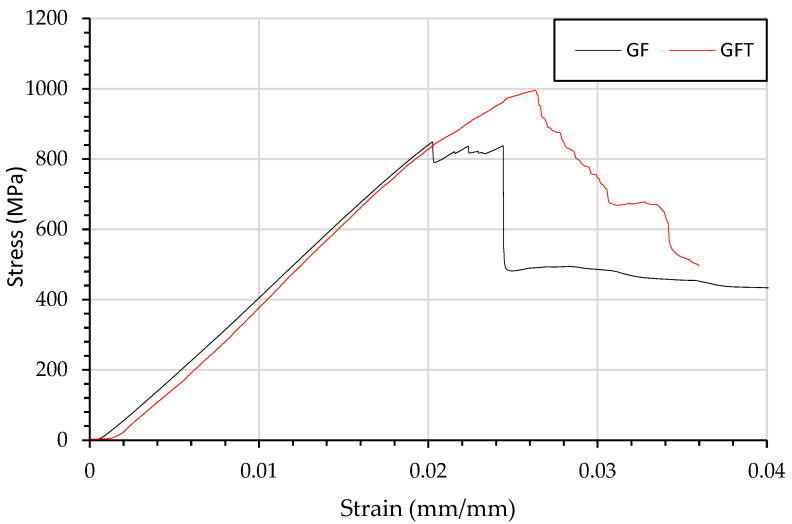
Flexural testing using three-point bending of composites beams and resulting stress strain curve of GF and GFT.

**Figure 7 materials-15-08877-f007:**
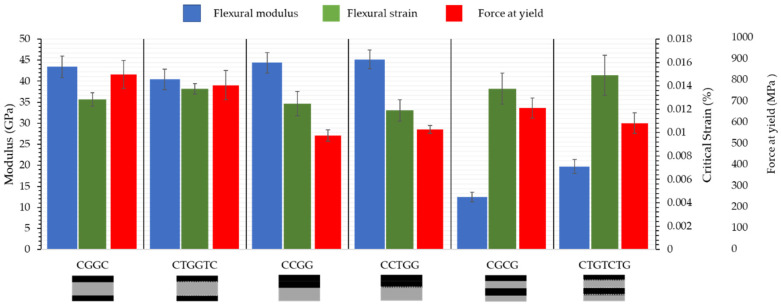
Flexural strength yield strength of hybrid fibres with carbon fibre under compressive loading and toughened hybrid fibres after three-point bending, showing a flexural modulus increase for alternating CTGTCTG. T denotes the interlayer veil toughening and number of veils used.

**Figure 8 materials-15-08877-f008:**
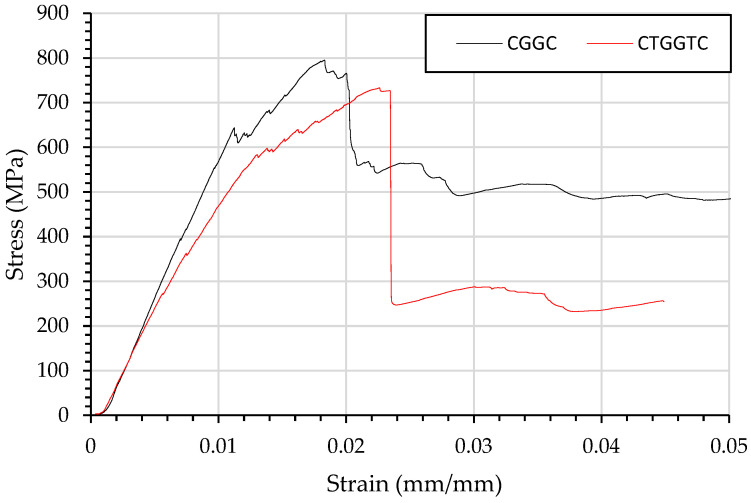
Flexural testing using three-point bending of composites beams and resulting stress strain curve of CGGC and CTGGTC.

**Figure 9 materials-15-08877-f009:**
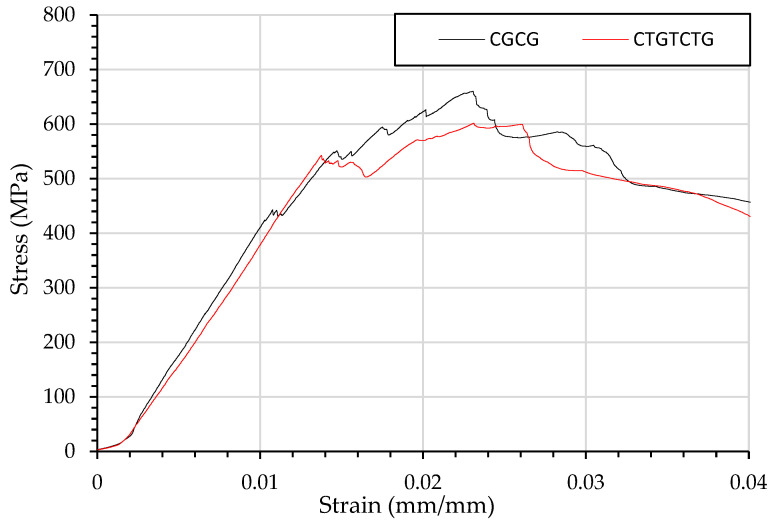
Flexural testing using three-point bending of composites beams and resulting stress strain curve of stress strain curve of CGCG and CTGTCTG.

**Figure 10 materials-15-08877-f010:**
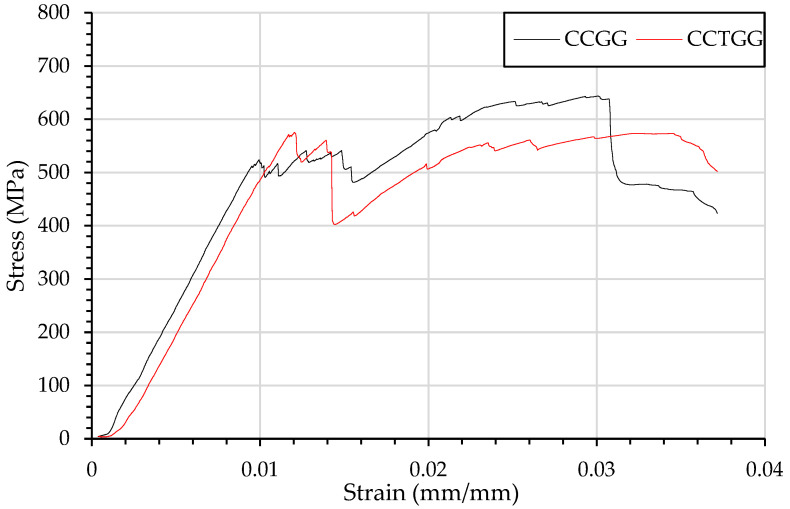
Flexural testing using three-point bending of composites beams and resulting stress strain curve of CCGG and CCTGG.

**Figure 11 materials-15-08877-f011:**
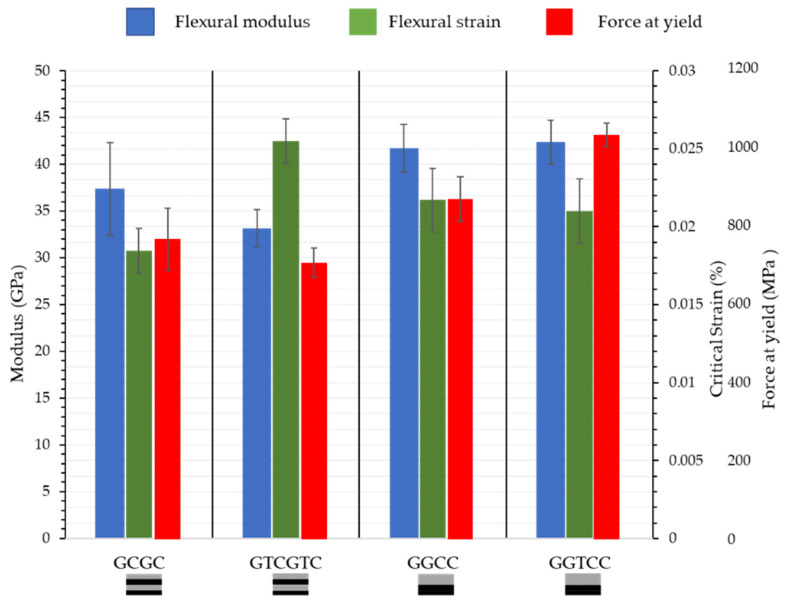
Flexural strength and modulus of hybrid fibre stacks with glass fibre undertaking compression stress after three-point bending. Showing a flexural increase for GGTCC and a strain increase for the alternating GTCTGTC sample. T denotes the interlayer veil toughening and the number of veils used.

**Figure 13 materials-15-08877-f013:**
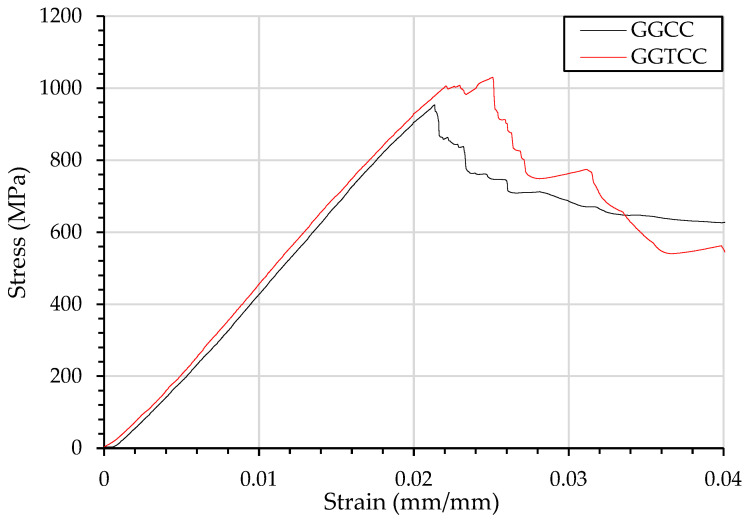
Flexural testing using three-point bending of composites beams and resulting stress strain curve of GGCC and GGTCC.

**Figure 14 materials-15-08877-f014:**
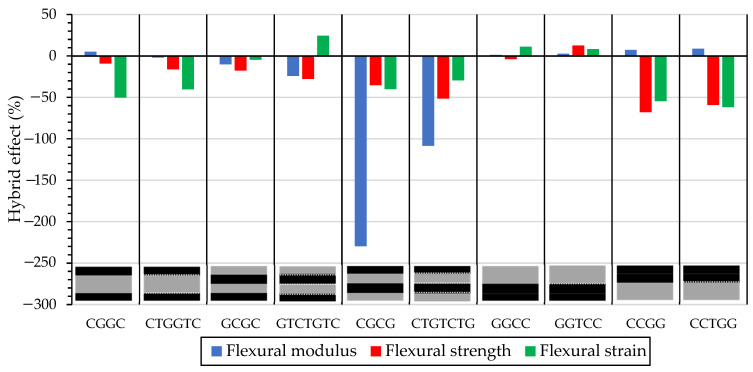
Rule of mixture comparison between hybrid stacking sequences and the veil toughened hybrid stacks.

**Figure 15 materials-15-08877-f015:**
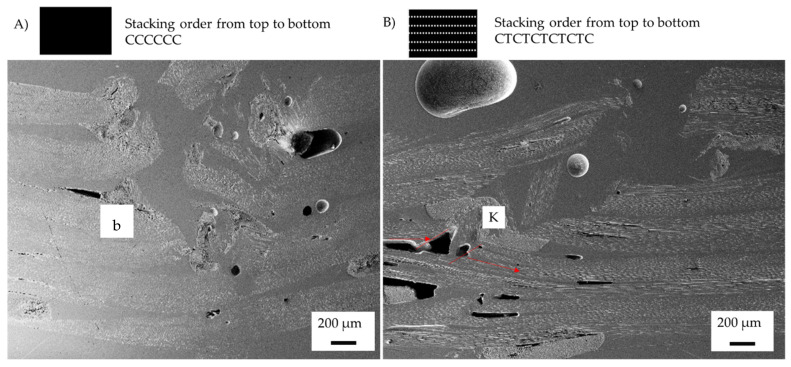
Post-failure micrographs of control samples after three-point bending and toughened samples with Xantu.Layr^®^: (**A**) control carbon fibre (CF), (**B**) carbon fibre toughened (CFT)Annotation: K (fibre kinking) and b (fibre buckling). Red lines indicate fibre failure.

**Figure 16 materials-15-08877-f016:**
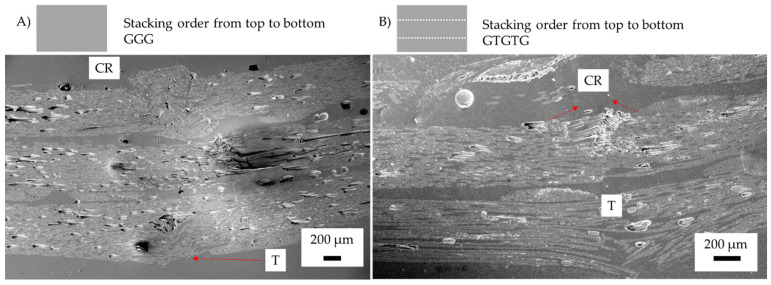
Post failure micrographs of control samples after three-point bending and toughened samples with Xantu.Layr^®^: (**A**) control glass fibre (GF), (**B**) glass fibre toughened (GFT), Annotation: CR (fibre crushing) and T (tension rupturing). Red lines indicate fibre failure.

**Figure 17 materials-15-08877-f017:**
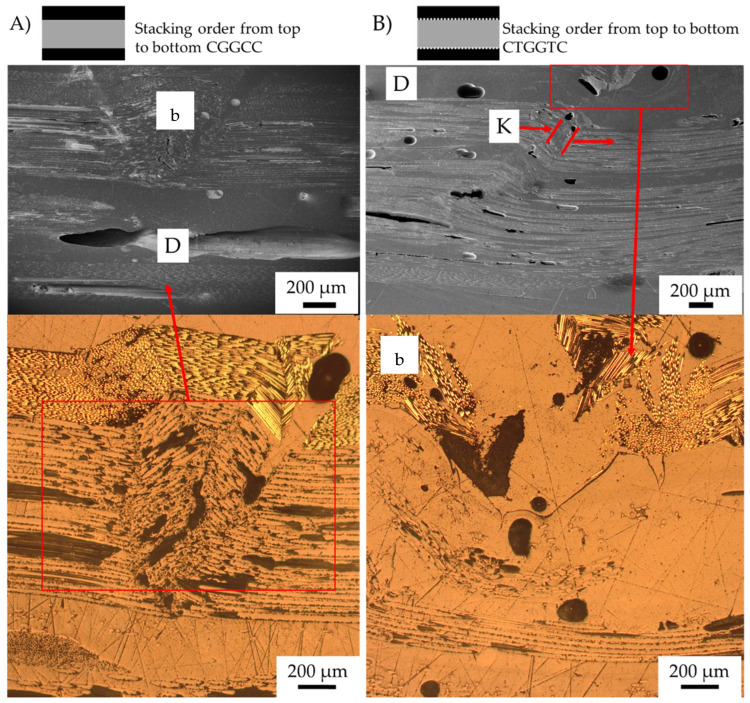
Post-failure micrographs of Carbon/Glass hybrid fibre samples after Three-point bending (**A**) CGGC (S) (**B**) CTGGTC. Annotation K (fibre kinking), b (fibre buckling) and D (Delamination). Red lines indicate fibre failure, fibre direction shown in with red arrow.

**Figure 18 materials-15-08877-f018:**
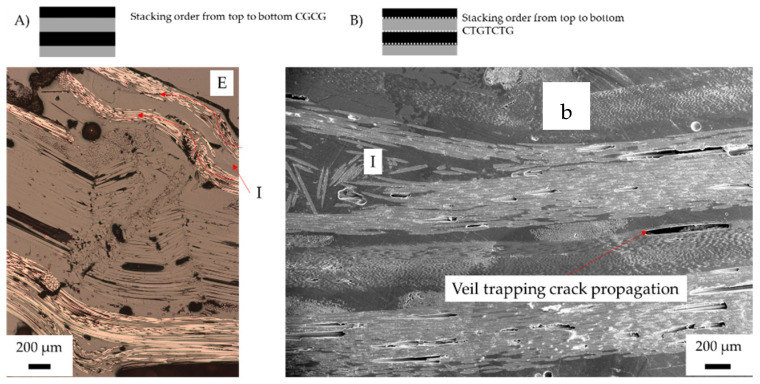
Post-failure micrographs of carbon/glass hybrid fibre samples after three-point bending: (**A**) alternating (CGCG) failure mechanism of toughened structures with carbon fibre under compression; (**B**) CTGTCTG. Annotation: E (elastic buckling), I (inter fibre failure) and b (fibre buckling). Red lines indicate fibre failure, fibre direction shown with red arrow.

**Figure 19 materials-15-08877-f019:**
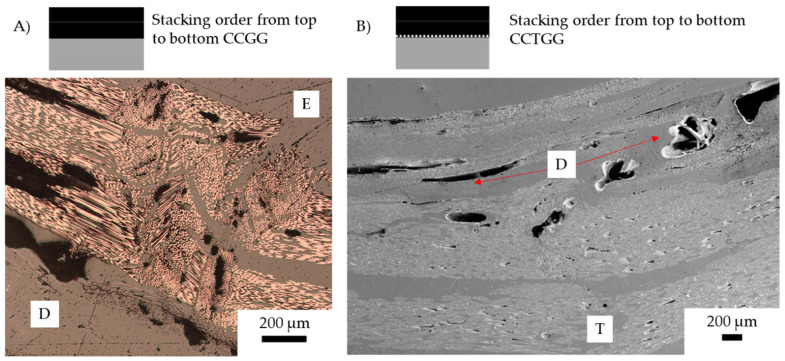
Post-failure micrographs of carbon/glass hybrid fibre samples after three-point bending: (**A**) CCGG, (**B**) CCTGG. Annotation: E (elastic buckling), T (tension rupturing), and D (Delamination). Fibre direction shown with red arrow.

**Figure 20 materials-15-08877-f020:**
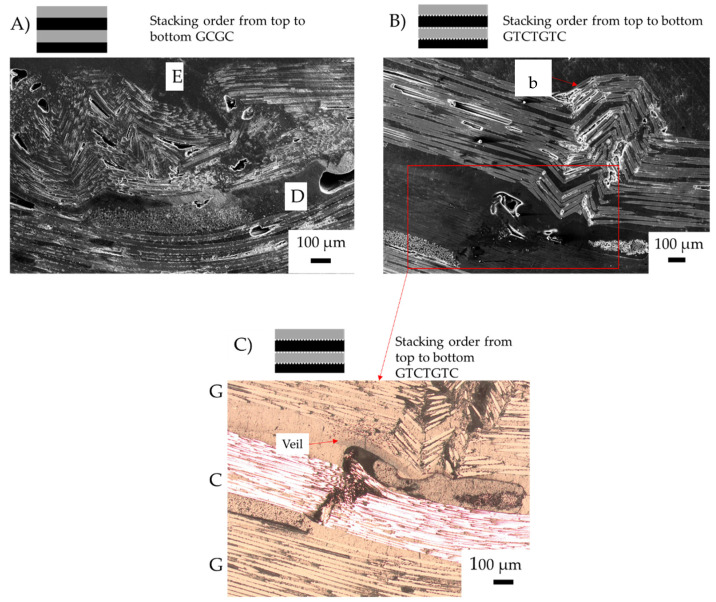
Post-failure micrographs of carbon/glass hybrid fibre failure mechanisms after three-point bending (**A**) GCGC; (**B**) GTCTGTC, failure mechanism of toughened structures with glass fibre under compression; (**C**) optical micrograph of veil trapping crack propagation in GTCTGTC notated by area D. Annotation: E (elastic buckling), b (fibre buckling) and D (delamination). Red lines indicate fibre failure, fibre direction shown with red arrow. Where G is the glass fibre ply and C is the carbon fibre ply.

**Figure 21 materials-15-08877-f021:**
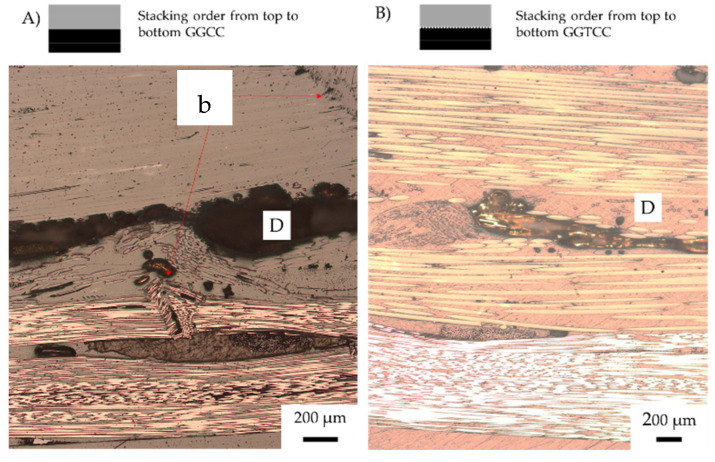
Post-failure micrographs of carbon/glass hybrid fibre failure mechanisms after three-point bending: (**A**) GGCC, (**B**) GGTCC. Annotation: b (fibre buckling) and D (delamination). Red lines indicate fibre failure, fibre direction shown with red arrow.

**Figure 22 materials-15-08877-f022:**
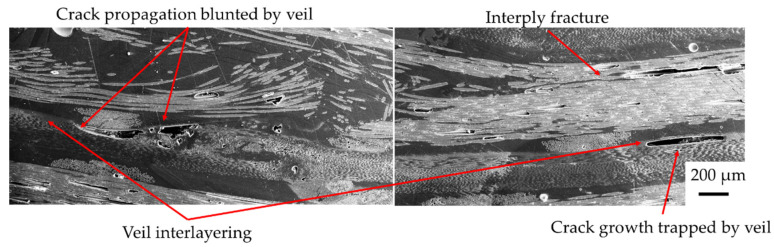
Veils effect of localisation of failure within the CTGTCTG, showing localisation leading to interplay failure preventing glass/carbon fibre interfacial delamination. Note this is the same sample with two images taken of different areas.

**Figure 23 materials-15-08877-f023:**
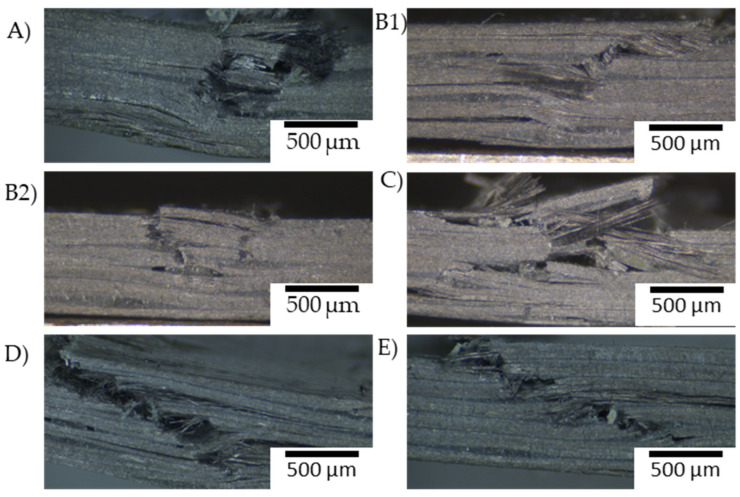
Analysis of crack propagation due to three-point bending test in carbon fibre interlayered with nanofibrous polyamide 6,6 veils: (**A**) No toughening; (**B**) Comparison of single-veil reinforcement of (**B1**) compressive and (**B2**) tensile area; (**C**) Comparison of reinforcement of two veils in compression; (**D**) Double veil reinforcement on the neural axis and (**E**) Fully interlayered carbon fibre stack with veil.

**Figure 24 materials-15-08877-f024:**
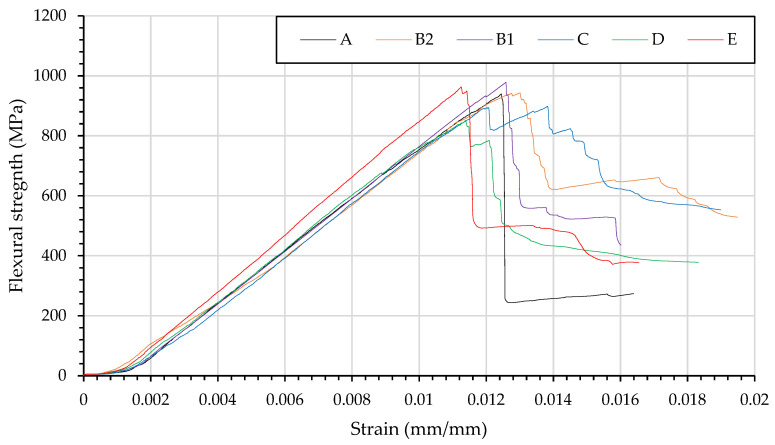
Stress strain curves of CFRP reinforced with PA 6,6 nanofibre (A) CF control sample; (B1) Single veil reinforcement compression layer of CFRP; (B2) Single veil reinforcement on tensile layer of CFRP; (C) Double compression layer of veil; (D) Double veil reinforcement on the neural axis and (E) Fully interlayered carbon fibre stack.

**Figure 25 materials-15-08877-f025:**
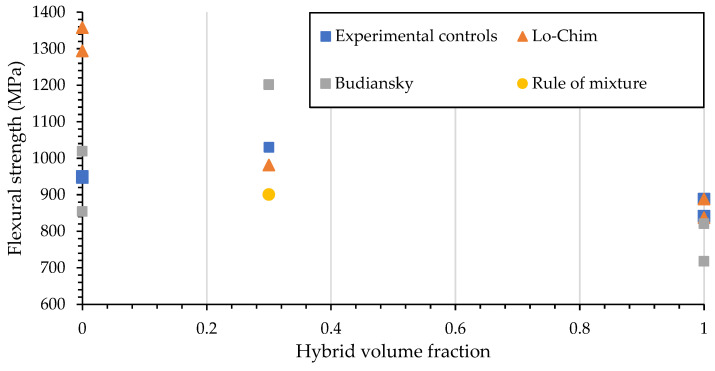
Flexural strength and hybrid volume fraction for mathematical derivations of the Lo-Chim and Budiansky models, compared with the experimentally produced CF, GF and GGCC samples and their veil toughened counterparts CFT, GFT and GGTCC.

**Table 1 materials-15-08877-t001:** Material Characterisation.

Material	Density (g/cm^3^)	Areal Density (g/m^2^)
PA 6,6 nanofibres	0.32	4.5
Glass fibre	2.60	850
Carbon fibre	1.80	300
Epoxy resin 270	1.15	NA

**Table 2 materials-15-08877-t002:** Stacking Sequence for Hybrid Fibre Samples.

Sample Name	Stack Illustration with White Lines Showing Veil Interlayering	Number of Fibre Plies	Number of Interlayers of Veil	Fibre Volume Fraction after Cure (%)
Control GF	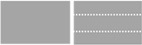	3	0	67.10
Control GF T		3	2	
Control CF	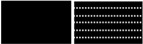	6	0	63.70
Control CF T		6	5	
CGGC	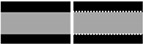	4	0	65.12
CTGGTC		4	2	
GCGC	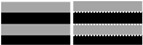	4	0	64.88
GTCTGTC		4	3	
CGCG	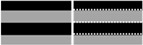	4	0	
CTGTCTG		4	3	
GGCC	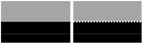	4	0	64.97
GGTCC		4	1	
CCGG	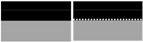	4	0	
CCTGG		4	1	

C = Carbon Fibre; G = Glass Fibre; T = Veil Interlayering.

**Table 3 materials-15-08877-t003:** Interlayering of carbon fibre stacks with nanofibrous polyamide 6,6 veils.

A	B1	B2	C	D	E
Control	Single veil reinforcement	Single veil reinforcement	Double compression layer veil	Double veil reinforcement on the neural axis	Fully interlayered carbon fibre stack
	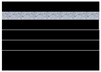	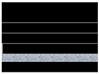	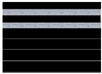	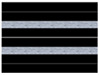	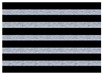

**Table 4 materials-15-08877-t004:** Flexural strength and modulus for interlayer toughened carbon fibre.

Sample Type	Flexural Modulus (GPa)	Flexural Strength (MPa)
A	54.46 ± 3.97	881.24 ± 114.11
B1	57.30 ± 4.41	924.71 ± 94.20
B2	55.95 ± 7.94	792.25 ± 61.97
C	50.553 ± 3.71	820.33 ± 39.88
D	47.20 ± 5.38	766.166 ± 19.93
E	58.86 ± 5.02	956.46 ± 52.55

## Data Availability

All data and code are available within the text.

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
