# Peer review of "Evaluation of the Failure Mechanism in Polyamide Nanofibre Veil Toughened Hybrid Carbon/Glass Fibre Composites"

_materials, 2022, doi:10.3390/ma15248877_

Round 1

Reviewer 1 Report

The manuscript having ID: Materials-2080769 discusses failure modes taking place in the hybrid FRPCs consist of glass fibers and glass fibers as reinforcement in epoxy under the three-point bending tests and Mode I testing. The manuscript is well-written though there are some minor mistakes in writing and presentation of some graphs. As far as the technical content is concerned, the contribution is worthy to be published. I recommend to publish this manuscript with the following modifications/responses:

1.               Refer the equation as for instance in line 168. Why there is again in line 171? Correct for all equation.

2.               Define each and every symbol to be used in the eqns.

3.               The graphs are not in a good presentable form. Enhance the quality. Follow any good article.

4.               Show some of the fractured and tested samples.

5.               Group figures like a, b ,c if they are presenting same type of results.

6.               Cite the following article which study the fracture of facesheet of FRPCs in three-point bending.

 https://doi.org/10.1016/j.compstruct.2022.116169

Author Response

Grammar and spelling check was performed throughout the article to improve the readability. The following changes have been made based on the reviewer’s input.

Best regards,

Ashley Blythe and co-authors

  1. Refer the equation as for instance in line 168. Why there is again in line 171? Correct for all equation and Define each and every symbol to be used in the eqns.

Equations updated to include all relevant details under each equation.

  1. The graphs are not in a good presentable form. Enhance the quality. Follow any good article.

Graphics now are updated to show all equation parts and symbol and only relevant failure mechanisms.

-Change dots from different colours to x square circle to aid clarity

-Removed text that wasn’t relevant to the graphics

-Colour change to highlight key areas

-Overall graphics were changed to be same size and orientation

  1. Show some of the fractured and tested samples.

The images of the samples that are not cut and polished are blurry due to the microscope being unable to focus on the high level of out of plane damage. Other samples are taken from a phone, and it was deemed they were not fit for publication.

  1. Group figures like a, b, c if they are presenting same type of results.

All figures are changed to ABC formatting.

  1. Cite the following article which study the fracture of facesheet of frpcs in three-point bending.

This reference (ref 38) is added into the manuscript, showcasing similar nanomaterials undergoing three-point bending research.

Reviewer 2 Report

In this work, the authors investigated the bending failure mechanisms within toughened interlayer hybrid composites. Also, the flexural modulus was optimized using stacking sequences of interlayered hybrid of carbon and glass fibre plies. The manuscript is well prepared; however, the following comments can help improve the manuscript:

·         The novelty of the work is not clearly highlighted.

·         Results and discussions have not been compared and verified to the findings of the other researchers.

·         In Figure 19, the authors need to review the Figure caption because some Annotations such as CR (fibre crushing), K (fibre kinking), I (inter fibre failure), and B (fibre buckling) are not belonging to Figure 19. Also, there is no any Red line in the Figure.

·         On page 2, paragraph 3, lines 71-85, the authors are recommended to review https://doi.org/10.1016/j.mechmat.2021.104025 to show how the mechanical properties of a hybrid carbon/glass fibre composite could be improved.

Author Response

Grammar and spelling check was performed throughout the article to improve the readability. The following changes have been made based on the reviewer’s input.

Best regards,

Ashley Blythe and co-authors

Reviewer 2

  1. The novelty of the work is not clearly highlighted.

The abstract is updated to clearly highlight the novelty of the project in the nanofibre stacking sequences.

  1. Results and discussions have not been compared and verified to the findings of the other researchers.

Discussion compared to other researchers and their calculated and mechanically tested values.

  1. In Figure 19, the authors need to review the Figure caption because some Annotations such as CR (fibre crushing), K (fibre kinking), I (inter fibre failure), and B (fibre buckling) are not belonging to Figure 19. Also, there is no any Red line in the Figure.

All graphics have been updated to include the relevant failure mechanisms.

  1. On page 2, paragraph 3, lines 71-85, the authors are recommended to review https://doi.org/10.1016/j.mechmat.2021.104025 to show how the mechanical properties of a hybrid carbon/glass fibre composite could be improved.

This was added as ref 33 to the last paragraph as it is relevant to fibre bridging mechanism for nanofibres and provides additional support to the mechanical behavior found within this paper.

Round 2

Reviewer 1 Report

The manuscript has been improved.

Thanks!